# 1,4-α-Glucosidase from *Fusarium solani* for Controllable Biosynthesis of Silver Nanoparticles and Their Multifunctional Applications

**DOI:** 10.3390/ijms24065865

**Published:** 2023-03-20

**Authors:** Ying-Jie Zeng, Xiao-Ling Wu, Hui-Rong Yang, Min-Hua Zong, Wen-Yong Lou

**Affiliations:** 1College of Food Science & Technology, Southwest Minzu University, Chengdu 610041, China; 2Laboratory of Applied Biocatalysis, School of Food Science and Engineering, South China University of Technology, No. 381 Wushan Road, Guangzhou 510640, China

**Keywords:** 1,4-α-glucosidase, AgNPs biosynthesis, multifunctional applications, *Fusarium solani* DO7

## Abstract

In the study, monodispersed silver nanoparticles (AgNPs) with an average diameter of 9.57 nm were efficiently and controllably biosynthesized by a reductase from *Fusarium solani* DO7 only in the presence of β-NADPH and polyvinyl pyrrolidone (PVP). The reductase responsible for AgNP formation in *F. solani* DO7 was further confirmed as 1,4-α-glucosidase. Meanwhile, based on the debate on the antibacterial mechanism of AgNPs, this study elucidated in further depth that antibacterial action of AgNPs was achieved by absorbing to the cell membrane and destabilizing the membrane, leading to cell death. Moreover, AgNPs could accelerate the catalytic reaction of 4−nitroaniline, and 86.9% of 4-nitroaniline was converted to p-phenylene diamine in only 20 min by AgNPs of controllable size and morphology. Our study highlights a simple, green, and cost-effective process for biosynthesizing AgNPs with uniform sizes and excellent antibacterial activity and catalytic reduction of 4-nitroaniline.

## 1. Introduction

Nanoparticles have had significant applications in many fields, including food, catalysis, energy conversion, environment, and biomedicine. [1,2,3,4,5,6]. Among the metallic nanoparticles, silver nanoparticles (AgNPs) are widely applied in plenty of different aspects, such as food preservation, water treatment, catalyst, batteries, cosmetics, and personal care products, due to their broad-spectrum antibacterial activities [7]. However, large numbers of the as-prepared AgNPs remain on the laboratory scale owing to their weak stability in real-world samples/environments, the sharp variation between batches, and their lack of multi-functionality, which finally gave rise to variations in the application performances [8]. To minimize or eliminate the above-mentioned issues, finding out the suitable capping agents at the nanoparticle surface is urgent and necessary. Meanwhile, the conventional synthesis of AgNPs by chemical and physical methods [9] usually involve harsh conditions, highly toxic reductants, and organic solvents, precluding their biomedical applications [10]. To make matters worse, the toxicity mainly caused by the preparation process of eukaryotic and mammalian systems severely limits the further use of AgNPs for antimicrobial therapies. Clearly, it is fundamentally essential to explore green synthesis strategies of AgNPs for fighting toxicity. Therefore, the green synthesis methods, including the use of plant extracts, biodegradable polymers, and enzymes/bacteria [11], have received the most extensive attention recently due to their status as environmentally friendly alternatives as well as their cost-effectiveness, easily scalable production processes, and extensive applications. Moreover, the green chemistry synthesis of AgNPs had been considered a safe therapeutic agent in animal models [12]. Due to the unique properties of AgNPs that are closely linked to their size and morphology, it has been necessary to control the size and morphology of nanoparticles [13,14]. Unfortunately, the size of AgNPs varies substantially depending on the used ligand and reducing agent, and AgNPs with sizes above 20 nm are not optimal for antimicrobial applications [11]. AgNPs (sizes ≥ 20 nm) primarily rely on the release of silver ions to exert antimicrobial efficacy [15], while smaller-sized AgNPs, especially those 10 nm and below, could directly interact with bacteria and then kill them [16]. Furthermore, the introduction of an effective stabilizer is a determining factor of the final size of metal nanoparticles. For example, the addition of stabilizers such as diethylenetriamine, polydopamine, polyvinylpyrrolidone (PVP), and polyethylene glycol (PEG) has been demonstrated to enhance the stability of nanoparticles system and prevent nanoparticle aggregation [17]. Interestingly, the stabilizers coated on AgNPs could reduce their toxicity to mammalian cells [18].

Water is essential for the living beings on the earth. Unfortunately, the quality of water resources has been continuously deteriorated by pollutants such as heavy metals and organic pollutants [19], which would inevitably bring harm to food production when used by food factories. Moreover, these pollutants possess serious side-effects and toxicities, and even a small amount of contact could be lethal and carcinogenic, such as the case of 4−nitroaniline (4−NA). Furthermore, some diseases might be attributed to the microbes present in wastewater, and these main detrimental microbes are bacteria [20]. Bacteria, especially *Escherichia coli* (*E. coli*), are considered as part of the pathogeny responsible for several illnesses called waterborne diseases. Interestingly, biogenic nanoparticles have been of immense interest in the reduction of environmental contaminants due to their green synthesis method and good performance [21]. Moreover, AgNPs have shown excellent catalytic performance for the reduction of 4-nitrophenol by NaBH_4_ in aqueous solutions [22].

It has been reported that some bioactive compounds from *Fusarium* species [23] could effectively synthesize AgNPs, but little research has focused on which compound in these strains was exactly responsible for AgNP synthesis. In this study, the biomolecule responsible for the conversion of Ag^+^ to form AgNPs in endophytic *F. solani* DO7 isolated from *Dendrobium officinale* was confirmed (Figure 1). The size and morphology of AgNPs were controlled via the addition of a stabilizer. Moreover, we also tried our best to reveal the antibacterial mechanism of AgNPs against *E. coli* because the exact mechanism of action remained somewhat elusive. Finally, the applications of AgNPs, including antimicrobial activity, catalytic activity, and kinetic properties, were also evaluated.

## 2. Results

### 2.1. Preparation of AgNPs

After the incubation and cell disruption of *F. solani* DO7, the cell lysate was found to synthesize AgNPs with the addition of 1 mM of AgNO_3_ at 25 °C and a pH of 7.0. The color changed from colorless to light brown within the first 24 h (Figure 1a) compared to the control group without AgNO_3_ addition. The reaction solutions were still exhibited as hydrosol with no obvious precipitation even after 96 h of incubation. Moreover, the formation of AgNPs was further confirmed by UV-Vis spectroscopy (Figure 1a). A specific peak at 423 nm, which corresponded to the characteristic surface plasmon resonance of AgNPs, could be observed after the formation of AgNPs. The plasmon band of the AgNPs synthesized in the cell lysate was symmetric, indicating that the solution did not generate many aggregated particles according to the occurrence of Mie scattering, and the results were consistent with the TEM images (Figure 1b).

### 2.2. Assay of the Compositions Responsible for AgNPs Synthesis

To find out which ingredient was responsible for the Ag^+^ to form AgNPs, the ammonium sulfate precipitation method was used to salt out the active proteins. The protein that was precipitated with 80% saturated ammonium sulfate solutions exhibited the best AgNP synthesis effect in the presence of coenzyme β-NADPH (Appendix A). The obtained protein was further purified using ion exchange chromatography, and the third collected fraction (F3, Appendix A) showed the excellent synthesis effect due to its narrow half-peak width. The SDS-PAGE analysis indicated that F3 was a protein with the molecular weight of 72.8 kDa (Figure 1c). The protein band at 72.8 kDa was excised and trypsinized prior to mass spectrometry sequencing. Four peptides, TDPDYWYTWTR, ATALTLYANWLVSHGDR, SAVHLTWSYASF VGAAER, and TGYDLWEEVNGSSFFTLSA SHR, were obtained and matched to glucan 1,4-α-glucosidase of *F. proliferatum* (accession number: CVL01304.1) in NCBIprot databases (Figure 1d). The coverage ratio of the protein sequence was 11%.

It is reported that Ag/BSA composite nanospheres could be synthesized from water-in-oil microemulsions containing aqueous solutions of AgNO_3_ and KBH_4_ using compressed CO_2_ as an anti-solvent [24]. However, no AgNPs were formed in the BSA solutions even in the presence of β-NADPH (Appendix A).

### 2.3. Controllable Synthesis of AgNPs

To obtain the accurate control of the nanoparticles’ morphology and size, PVP stabilizers were introduced to the reaction system. After adding different masses of PVP to the AgNP synthesis system, nanospheres of uniform size were observed by TEM (Figure 2a). The addition of PVP contributed to improving the dispersity of AgNPs and to preventing the particles’ agglomeration, whereas the improvement effect was strongly associated with the mass ratio of Ag^+^ and PVP. When the mass ratio of Ag^+^/PVP was too low (1:0.5) or high (1:2), there was nanoparticle aggregation. Only under the 1:1 mass ratio of Ag^+^/PVP conditions, the synthesized AgNPs were uniform in size and morphology (Figure 2a). Moreover, the characteristic surface plasmon resonance of AgNPs was still located near 420 nm, indicating that there was no inhibitory effect of PVP on nanoparticle synthesis (Figure 2b). However, when the mass ratio of Ag^+^/PVP changed, the width of and shift in the AgNP characteristic peak varied (Figure 2b). In addition, when the pH value became neutral, the solution changed from colorless to dark yellow in 24 h. In contrast, when the medium changed from neutral to alkaline, for example at pH 10.0, the time of the color change decreased to 12 h. AgNPs of uniform size were obtained under the above conditions, and the size of nanoparticles showed a lognormal size distribution (Figure 2a,c). Under the optimum pH conditions, the average size of the AgNPs was 9.57 ± 0.36 nm.

### 2.4. Characteristics of AgNPs

The interaction between PVP and AgNPs was further identified by FTIR measurements (Appendix A). The strong broad peak at 3300~3500 cm^−1^ was characteristic of the N–H stretching vibration of AgNPs, and it was overlapped with the O-H stretching vibration of PVP. The band at 2510 cm^−1^ was attributed to the -SH stretch vibration because the protein structure might have been damaged under alkaline pH conditions. Moreover, the disappearance of the -SH stretch vibration indicated the formation of a of Ag–S–PVP bond after the addition of PVP. Notably, the maximum absorbance observed at 1650 cm^−1^ was ascribed to the C=O stretch vibration in the amide linkages between PVP and AgNPs. The band at 1585 cm^−1^ indicated another amide band, and this was due to the combination of the N–H bending mode and C–N stretch vibration mode [25]. In addition to amide bonding, the band at 1440 cm^−1^ was identified as the methylene scissoring vibrations. The bands at 1048 cm^−1^ could be the C-N stretching vibrations of aliphatic amines, and they overlapped with the absorption peaks of C–O–C or C–O in PVP. Obviously, there was no significant absorption peak change in the AgNPs in the presence of PVP, and it is suggested that the coating surfaces of the PVP contributed to a capping effect on the AgNPs.

The presence of AgNPs was further confirmed by XRD. The Braggs reflections in the XRD pattern were observed at 2θ = 26.4°, 30.6°, 34.7°, 46.1°, 65.2°, and 78.3°, as shown in Appendix A. Obviously, there was a strong diffraction peak located at 34.7°, which was ascribed to the (111) facets of Ag. Additionally, the main peaks at 46.1°, 65.2°, and 78.3° (2θ values) corresponded to the (200), (220) and (311) planes, respectively. The above results further confirm the specific crystalline phase of the as-synthesized AgNPs [26]. An energy dispersive spectrometer further demonstrated the existence of silver (Appendix A).

### 2.5. Antibacterial Mechanism of AgNPs

The particle size and morphology of AgNPs has been reported to be closely related to their antibacterial activity. Therefore, the MICs and MBCs were determined (Figure 3a,b). The MICs of AgNPs against two strains were from 9 to 64 μg/mL. However, the MICs of mAgNPs (AgNPs synthesized with a PVP addition) were decreased nearly 2-fold to 9 µg/mL, indicating that the uniformity and stability of the nanoparticles were beneficial to inhibiting the growth of bacteria. Besides, the control groups (cell lysate, PVP, and 1,4-α-glucosidase) showed no antibacterial effect on the tested strains. Moreover, mAgNPs could significantly reduce the MBC by nearly two-fold to 20.3 µg/mL.

In view of the excellent antibacterial performance against *E. coli*, the action mechanism was further studied based on the previous study that showed that AgNPs attaching to a cell membrane interrupted the functions of the cell membrane such as its permeability and respiration [27]. When AgNPs made contact with the cell walls of *E. coli*, the Ag^+^ released from the AgNPs on the cell walls could cause changes in membrane potential and further disrupt membrane integrity (Figure 4). Moreover, under the stress of AgNPs, the intracellular ROS levels of *E. coli* increased, the MDA contents increased, the membrane permeability increased, the membrane potential decreased, and the activities of Na^+^-K^+^-ATPase, and Ca^2+^-Mg^2+^-ATPase showed a compensatory increase first and then decreased (Figure 3c).

### 2.6. Catalytic Reduction of 4−NA

The as-prepared AgNPs as catalysts in the reduction of 4−NA by NaBH_4_ were investigated. Distilled water was taken as the blank group, 1,4-α-glucosidase was used for the negative group, and the reaction system in the blank and negative groups also included 4-NA and NaBH_4_ (Figure 5). The 4-NA reduction process was tracked by detecting the variation of absorbance values within 120 min (Figure 5 and Appendix A). 4−NA was transformed to p-phenylene diamine during the reaction (Figure 5a). The intensity of the absorbance band at 380 nm (the characteristic absorption peak of 4−NA) decreased (Figure 5b), while a new absorption peak at 308 nm (the characterized absorption band for p-phenylene diamine) appeared, and its intensity enhanced with the increase in the reaction time. The above results confirmed that the p-phenylene diamine formed [28]. Furthermore, in the blank group, the absorbance at 380 nm was only marginally decreased by time without AgNP addition (Figure 5b and Appendix A), and it suggested that only a little amount of 4−NA was converted to p-phenylene diamines in the presence of NaBH_4_. The effect of 1,4-α-glucosidase in the reaction was weak (Appendix A). However, the addition of AgNPs and mAgNPs could considerably accelerate the reaction of 4−NA due to the fast reduction of the absorbance within the 2 h reaction time (Figure 5b and Appendix A), which indicated that the AgNPs and mAgNPs exhibited good catalytic action on the conversion of 4−NA. Compared to the AgNPs, the mAgNPs of controllable size and morphology showed a faster catalytic effect. In the case of the AgNPs, the conversion efficiency of 4−NA to p-phenylene diamine changed from 51.6% at 20 min to 97.6% at 120 min.

The kinetics of the catalytic reaction of 4−NA by NaBH_4_ with AgNPs were studied based on the pseudo-first order model, and the results are shown in Table 1. The rate constant of the reaction was markedly enhanced by tuning the size of nanoparticles in reaction system. The k value varied from 2.3 × 10^−3^ m^−1^ to 12.4–35.6 × 10^−3^ m^−1^ after the addition of nanoparticles. The increase in the k value further confirmed the catalytic activation of the nanoparticles in the reaction system. The catalytic performance of the mAgNPs was found to be much better than that of the AgNPs, as the *k* for the mAgNPs (35.6 × 10^−3^ min^−1^) was nearly three-fold higher than that of the AgNPs (12.4 × 10^−3^ min^−1^), which suggested that the increase in particle size had a negative effect on catalytic performance. Furthermore, the halfway point of the reaction time presented a negative correlation with the rate constant value because it sharply dropped after the nanoparticles were added. The values of t_1/2_ were markedly decreased from 263.7 min to 19.5~56.3 min when the AgNPs or mAgNPs were used as the catalyst. The results summarize that 10 mg of the AgNPs could be efficient in the reduction of 122 mg/L 4−NA in 20 min by using 15 mM NaBH_4_.

## 3. Discussion

The characteristic absorption spectrum of AgNPs appeared between 390 and 420 nm due to Mie scattering [29], responding to the silver metal. Therefore, an absorption peak appearance around 423 nm indicated that the AgNPs were formed. Besides, the plasmon bands were broadened with a tail in the longer wavelengths due to the non-uniform size distribution of the nanoparticles. Since the intensity of the plasmon resonance depended on the cluster sizes, the number of particles could not be linearly related to the absorbance intensities [30]. After the analysis of the compositions responsible for AgNP synthesis, the protein F3 was identified as 1,4-α-glucosidase. Moreover, the purified enzyme could synthesize AgNPs (Figure 1e) of better size and morphology than the cell lysate (Figure 1a). Moreover, some amino acids in the protein sometimes had weak reduction capacities, and bovine serum albumin (BSA) with an abundance of amino acids was applied to synthesize AgNPs. No AgNPs being formed in the BSA solutions (Appendix A) further suggested that 1,4-α-glucosidase from the cell lysate was supposed to be the enzyme with redox activity. Meanwhile, the synthesis mechanism investigation showed that the 1,4-α-glucosidase led to the Ag^+^ reduction and the AgNPs formation in the presence of β-NADPH. Moreover, it further indicated that 1,4-α-glucosidase served as a reducing agent in the AgNP synthesis process.

Although PVP had been reported as a protective agent which played a decisive part in controlling nanoparticle size in chemical synthesis [31], there has been little research on applying PVP during the formation of biosynthetic nanoparticles. However, in our study, a noticeable variation was observed when the Ag^+^/PVP mass ratio was 1:1. Based on the Mie scattering, the narrow absorption peak and short absorption tail indicated the narrow size distribution, and it corresponded to the TEM results (Figure 2a). Meanwhile, the TEM images also suggested that PVP contributed to improving the dispersity of AgNPs and to preventing the particles’ agglomeration. Importantly, particle size decreased when PVP was added. The coordination residues on the PVP chains had a physical bond with the nitro groups in the AgNPs, which enhanced the electronic cloud density of Ag^+^, the formation of the Ag^+^ crystal nucleus, and the transformation to Ag of Ag^+^ [32]. The interaction between PVP and Ag^+^ facilitated the formation of AgNPs as indicated by a color change. Therefore, the brown color changes of the AgNP solutions in the presence of PVP appeared at an earlier time, suggesting that PVP had a promoting effect on AgNP formation. Additionally, the half-peak width became narrower initially and then increased when the mass ratio of Ag^+^ and PVP changed from 1:0.5 to 1:2 (Figure 2a), further indicating that the interaction between PVP and AgNPs according to the Mie scattering and the synthesis process was not intervened by PVP. Furthermore, PVP was used as a stabilizer in our system, and it might have had a steric hindrance effect that limited the growth of the crystal surface of the AgNPs [33]. The fabrication of AgNPs of controllable size required the exact match bewteen the reaction rate and the mass transfer rate. Therefore, when excess PVP was applied, the long chain of PVP became entangled and then resulted in unstable AgNPs with slight aggregation (Figure 2a). In contrast, if the content of PVP was too low to coat over the crystal nucleus of the AgNPs, it would lead to silver ion collision and aggregation (Figure 2b). Therefore, the UV-Vis spectroscopy, TEM images, and particle size distribution together suggested that the optimal mass rate of Ag^+^ to PVP producing AgNPs of uniform size and morphology was 1:1. The presence of different functional groups, such as C-N and C-O-C, further demonstrated the interactions between silver ions, the protein, and PVP [34]. Meanwhile, results have been reported that are similar to our finding that the above functional groups originating from 1,4-α-glucosidase and PVP served as the capping ligands and as the protecting coat of the nanoparticles [35].

In the analysis of antimicrobial mechanisms, under the stress of AgNP conditions, the high intracellular ROS in *E. coli* reacted with membrane unsaturated fatty acids by peroxidation, further reduced cell membrane fatty acid saturation, and finally accumulated MDA in cells. The increased MDA further caused damage to the cell membrane, reduced ATPase activity, and eventually resulted in the inactivation or apoptosis of *E. coli* (Figure 4 and Appendix A). Additionally, mAgNPs possessed better anti-bacterial activity than AgNPs did, and this also confirmed that the antibacterial activity was presented in a size-dependent manner [9]. The presence of PVP might be helpful to enhance the adhesion of these Ag^+^ ion nanocarriers to the microbial cell surface [36]. Meanwhile, the SEM images of *E. coli* further confirmed that the AgNPs could cause some changes in the cell morphology of *E. coli*, especially in cells treated with a high dose of AgNPs (Appendix A). The *E. coli* treated with a high dose of AgNPs adhered to each other due to the strong damage of cell membranes. These results further verified that AgNPs could achieve the sterilization actions via the interference with essential bacterial cell functions upon contact [37].

The conversion efficiency of 4−NA to a p-phenylene diamine reached up to 86.9% after only 20 min for the mAgNPs (Figure 5d), and this indicates that the size had a significant effect on the catalytic reaction of p-nitroaniline and that bigger size particles possessed a much lower reduction efficiency. The results suggested that AgNPs of controllable size and morphology could strongly accelerate the reaction of 4−NA and that it only needed 20 min to catalyze 86.9% of 4−NA to a p-phenylene diamine. Moreover, based on the results, the catalytic mechanism could be presented as follows: BH_4_^−^ with a rich electron acted as an electron-donating reagent and hydrogen supplier. Thus, the potential difference existing between the donor (NaBH_4_) and the acceptor (4−NA) ensured that the reduction reaction could easily occurred [38]. Additionally, 4−NA would adsorb on the surface of AgNPs as soon as the AgNPs appeared in the reaction system. Then, the catalytic reaction started by facilitating the electron transfer from BH_4_^−^ to 4−NA. Furthermore, BH_4_^−^ could also transport hydrogen species to AgNPs, and then promptly transformed to 4−NA.

## 4. Materials and Methods

### 4.1. Chemicals

A potato dextrose broth (PDB), potato dextrose agar (PDA), a nutritional agar medium (NA), silver nitrate (AgNO_3_), polyvinyl pyrrolidone k30 (PVP k30), a Mueller–Hinton broth (MHB), 4-Nitroaniline, sodium borohydride, propidium iodide (PI; Shanghai Yuanye Biotechnology Company, Ltd., Shanghai, China), a reactive oxygen species (ROS) assay kit (Beyotime Biotechnology Company, Ltd., Shanghai, China), a 2-[6-amino-3-imino-3H-xanthen-9-yl]-benzoic acid methyl ester (RH123, Sigma, Shanghai, China), a lipid peroxidation malondialdehyde (MDA) assay kit (Beyotime Biotechnology Company, Ltd., Shanghai, China), a Na^+^K^+^-ATPase assay kit, a Ca^2+^Mg^2+^-ATPase assay kit (Nanjing Jiancheng Bioengineering Institute, Nanjing, China), glutaraldehyde, and paraformaldehyde were used.

### 4.2. Preparation of AgNPs

AgNPs were synthesized according to the recipe described in the literature with some modifications [39]. Briefly, the fungus strain was incubated aerobically in a 200 mL PDB medium (pH 6.5) at 26 °C and 120 rpm [40]. After 7 d of incubation, the fungal mycelia were collected and washed thoroughly. The cell lysate was obtained and used for the synthesis of AgNPs with the addition of AgNO_3_ (the final concentration was 1 mM) at 25 °C in the dark. After the reaction, 1 mL of the sample was withdrawn, and the absorbance was measured using a UV-visible spectrophotometer (UV-Vis).

The resulting cell lysate was centrifuged at 6000× *g* and 4 °C for 10 min to obtain a cell-free supernatant. Then, the ammonium sulfate was added to the supernatant, and the protein deposit was collected and dialyzed. After dialysis, the crude enzyme solutions were purified. The elution fractions were collected and then concentrated with an ultra-filtration membrane (3 kDa, Millipore, Billerica, MA, USA). The purified protein fractions were used to synthesize AgNPs.

### 4.3. Isolation and Identification of Protein

The purified proteins were analyzed by sodium dodecyl sulfate polyacrylamide gel electrophoresis (SDS-PAGE) [41]. After electrophoresis, the bands were removed from the gel and digested by trypsin. After freeze-drying, the protein sample was mixed with a matrix solution. Mass spectra were recorded on a matrix-assisted laser desorption/ionization time-of-flight mass spectrometer (MALDI-TOF-MS, Bruker Daltonics, Karlsruhe, Germany), which was equipped with a nitrogen laser operated at 337 nm in linear mode. The fragment spectra were compared with those of the NCBI protein database using BLAST.

### 4.4. Characteristics of AgNPs

The formation of AgNPs was monitored by UV-Vis (Shimadzu Corporation, Kyoto, Japan) spectroscopy, which was conducted between the wavelengths 200 and 800 nm. Transmission electron microscopy (TEM, HT-7700; Hitachi Ltd., Tokyo, Japan) was used to observe the morphology of synthesized nanoparticles [42]. FTIR spectroscopy was applied to evaluate the functional groups in the formed nanoparticles at a resolution of 4 cm^−1^. The structure and composition of the AgNPs were detected by a D8 X-ray diffractometer (Bruker Biosciences Corporation, Billerica, MA, USA).

### 4.5. Size and Shape Control of AgNPs

The impact of the mass ratios of Ag^+^ and PVP K30 (1:0.5, 1:1, and 1:2) was evaluated by a UV-visible spectrophotometer and finally confirmed by TEM. During the experiment, the concentrations of AgNO_3_ were fixed at 1 mM, and the mass ratio varied with the weight of the PVP K30 added to the reaction system.

### 4.6. Antibacterial Activity and Mechanism Assay

The antibacterial activity of the AgNPs was evaluated against 2 tested strains, *E. coli* (Gram-negative) and *Staphylococcus aureus* (*S. aureus*, Gram-positive), by the agar disk diffusion method [43].

To further evaluate the antibacterial activity of the AgNPs, both the minimum inhibitory concentration (MIC) and the minimum bactericidal concentration (MBC) were determined according to descriptions in the previous literature [44]. The sample concentrations were set to 128, 64, 32, 16, 8, 4, 2, 1, 0.5, 0.25, and 0.125 μg/mL.

The antibacterial mechanism was explored by the analysis of membrane integrity, intracellular ROS accumulation, cell membrane potential (ΔΨm), malondialdehyde (MDA), and the activities of Na^+^K^+^-ATPase and Ca^2+^Mg^2+^-ATPase based on the descriptions of the previous literature [45].

### 4.7. Cell Morphology Observation

The *E. coli* cells were fixed with 2.5% (*v*/*v*) glutaraldehyde and 4% (*m*/*v*) paraformaldehyde solutions for 3 h at 25 °C, and then rinsed in PBS for three times. The rinsed cells were dehydrated in a series of ethanol concentrations (30, 40, 50, 60, 70, 80, 90, and 100%) and dried to the critical point (Samdri-PVT-3D). The cells were coated with gold (MSP-2S) and examined using a scanning electron microscope (SEM) (Hitachi S-3400N, Tokyo, Japan).

### 4.8. Catalytic Reduction Assay

The catalytic efficiency of the AgNPs was evaluated toward 4-NA. The catalytic reaction was performed as follows: 10 mg of the AgNPs was accurately weighed and transferred to a 25 mL serum bottle with 16 mL distilled water; 2 mL of 4-NA (2 mM) and 2 mL of NaBH_4_ (150 mM) were supplemented to this mixture system; and the absorbance of the above-mixture was immediately determined after the NaBH_4_ addition using a spectrophotometer until the reaction occurred for 120 min.

## 5. Conclusions

This study confirmed that 1,4-α-glucosidase from *F. solani* DO7 was responsible for the synthesis of monodispersed AgNPs in the presence of β-NADPH and PVP at a pH of 10.0. Furthermore, the AgNPs exhibited good antimicrobial activity and catalytic activity. Additionally, the study indicated that the antibacterial action of the AgNPs was achieved by their adherence to the cell membrane, destabilizing the membrane, finally leading to cell death. Meanwhile, the AgNPs could catalyze 89% of the 4-NA conversion to a p-phenylene diamine in 20 min. Our study highlights a green and cost-effective process of biosynthesizing AgNPs with antimicrobial and catalytic properties, which will have great potential in the food industry, and the agricultural and environmental protection.

## Data Availability

Data are contained within the article or the Appendix A.

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
