# Peer review of "1,4-α-Glucosidase from Fusarium solani for Controllable Biosynthesis of Silver Nanoparticles and Their Multifunctional Applications"

_ijms, 2023, doi:10.3390/ijms24065865_

Round 1
Reviewer 1 Report
In this paper, the biosynthesis of silver nanoparticles and the antibacterial and catalytic properties of silver nanoparticles are demonstrated. The corresponding experimental results provide relatively good support for these conclusions. But this paper needs major revisions to be accepted.
1. The English language of this whole paper needs to be revised.
2. The catalytic property of silver nanoparticles is discussed in the Results and Discussion section, but it is stated that silver nanoparticles have a reducing effect in the Abstract. What exactly is it?
3. How to deduce that the antibacterial mechanism is that silver nanoparticles lead to the destruction of membrane integrity by adhering to cell membrane?
4. How many particles are there in total in Figure 2c?
Author Response
Dear Reviewer,
We are very grateful to you for your consideration of our manuscript (Manuscript ID: ijms-2215846) for publication in International Journal of Molecular Sciences and helpful suggestions for revision. We have revised the manuscript carefully according to the reviewers’ comments and the editor’s suggestions. The issues raised by the reviewers have been addressed as follows. Page, line and figure numbers refer to the revised manuscript.
Response to the first Reviewer’s comments
In this paper, the biosynthesis of silver nanoparticles and the antibacterial and catalytic properties of silver nanoparticles are demonstrated. The corresponding experimental results provide relatively good support for these conclusions. But this paper needs major revisions to be accepted.
- The English language of this whole paper needs to be revised.
Response: Thanks for the referee's helpful suggestion. We are very grateful to the reviewer for pointing out the lack of the original manuscript. We have checked the manuscript carefully and then corrected the errors that the reviewer’s mentioned. Before we submit the revised manuscript, the English in this manuscript has been checked and modified by a native English speaker.
- The catalytic property of silver nanoparticles is discussed in the Results and Discussion section, but it is stated that silver nanoparticles have a reducing effect in the Abstract. What exactly is it?
Response: Thanks for the referee's helpful suggestion. We are very grateful to the reviewer for pointing out the lack of the original manuscript. We have modified the expression in the Abstract section as catalytic property.
- How to deduce that the antibacterial mechanism is that silver nanoparticles lead to the destruction of membrane integrity by adhering to cell membrane?
Response: Thanks for the referee's helpful suggestion. The antibacterial mechanism in our manuscript was deduced based on our research finding combined with previous literatures.
- How many particles are there in total in Figure 2c?
Response: Thanks for the referee's helpful suggestion. There are 874 particles in total in Figure 2c.
In addition, a number of typographical and language usage errors we noticed whilst revising the manuscript has been corrected. All changes are marked in red fonts throughout the revised manuscript.
We look forward to hearing from you again.
With kind regards,
Prof. Dr. Wen-Yong Lou
Lab of Applied Biocatalysis,
South China University of Technology,
Guangzhou 510640, China
Tel/Fax: +86-20-22236669
E-mail: wylou@scut.edu.cn (Prof. Lou)
Reviewer 2 Report
In this work, the authors investigated an environmentally friendly process for the synthesis of silver nanoparticles (AgNPs) based on the use of 1,4,-α-glucosidase from F. solani DO7 as ligand and reducing agent in the presence of β-NADPH as cofactor and PVP as stabilizer. The biomolecule responsible for the reduction of Ag+ to form AgNPs was isolated from the cell lysate, purified and identified by MALDI-TOF analysis. To determine the influence of the stabiliser on their properties, AgNPs were synthesised in the presence of different amounts of PVP. Finally, both the antibacterial activity of the obtained AgNPs and their catalytic activity for the reduction of 4-nitrophenol by NaBH4 in aqueous solution were determined.
The biosynthesis of metal nanoparticles is a topic of interest to a wide audience, and the article presents interesting results, which is why, in my opinion, it is worthy of publication. However, there are a few points that the authors need to consider:
- Pag 2 row 75. The authors should use the term “stabiliser” as PVP cannot be defined as a surfactant;
- Pag 3. Some details of the particle synthesis process are not clear: what is the amount of organic material, both in terms of cell lysate and purified protein, used in the AgNPs synthesis? What is the amount of coenzyme β-NADPH added in the case of the purified 1,4,-α-glucosidase? With regard to the synthesis in the presence of PVP, the authors refer to the mass ratio AgNO3/PVP in the experimental part, but to the mass ratio Ag+/PVP in the discussion. Please clarify.
- The fact that the addition of PVP allows the production of particles with smaller diameters and a more controlled particle size distribution was emphasized during the discussion. However, only one AgNPs sample was reported for the average size and relative particle size distribution histogram. Authors should provide this information for all samples prepared, at least in the Supporting Info section of the manuscript.
- Pag 6 row 252. The comment refers to Figure 2a.
- Pag 9 row 329. “Distilled water without any addition was taken as the blank” this sentence is unclear: I hope NaBH4 has been added to the blank. Please clarify.
- In the first part of the manuscript, until page 9, particles obtained in the presence of PVP as stabiliser were called mAgNPs, whereas in the second part they were called AgNPsM. Authors should use consistent nomenclature.
- Pag 10 rows 384-388. This paragraph should be deleted.
- There are some typographical and grammatical errors in the text. I advise the authors to check the manuscript carefully.
Author Response
Dear Reviewer,
We are very grateful to you for your consideration of our manuscript (Manuscript ID: ijms-2215846) for publication in International Journal of Molecular Sciences and helpful suggestions for revision. We have revised the manuscript carefully according to the reviewers’ comments and the editor’s suggestions. The issues raised by the reviewers have been addressed as follows. Page, line and figure numbers refer to the revised manuscript.
Response to the second Reviewer’s comments
In this work, the authors investigated an environmentally friendly process for the synthesis of silver nanoparticles (AgNPs) based on the use of 1,4-α-glucosidase from F. solani DO7 as ligand and reducing agent in the presence of β-NADPH as cofactor and PVP as stabilizer. The biomolecule responsible for the reduction of Ag+ to form AgNPs was isolated from the cell lysate, purified and identified by MALDI-TOF analysis. To determine the influence of the stabiliser on their properties, AgNPs were synthesised in the presence of different amounts of PVP. Finally, both the antibacterial activity of the obtained AgNPs and their catalytic activity for the reduction of 4-nitrophenol by NaBH4 in aqueous solution were determined.
The biosynthesis of metal nanoparticles is a topic of interest to a wide audience, and the article presents interesting results, which is why, in my opinion, it is worthy of publication. However, there are a few points that the authors need to consider:
- Pag 2 row 75. The authors should use the term “stabiliser” as PVP cannot be defined as a surfactant;
Response: Thanks for the referee's helpful suggestion. As suggested by the referee, we have modified the expression in the Page 2 row 75.
- Pag 3. Some details of the particle synthesis process are not clear: what is the amount of organic material, both in terms of cell lysate and purified protein, used in the AgNPs synthesis? What is the amount of coenzyme β-NADPH added in the case of the purified 1,4-α-glucosidase? With regard to the synthesis in the presence of PVP, the authors refer to the mass ratio AgNO3/PVP in the experimental part, but to the mass ratio Ag+/PVP in the discussion. Please clarify.
Response: Thanks for the referee's helpful suggestion. The levels of cell lysate and purified protein contained in the reaction system were 15 ug/mL (crude protein in cell lysate) and 1ug/mL, respectively. The amount of coenzyme β-NADPH added in the case of the purified 1,4-α-glucosidase was 2 U (200 U/mL, 10uL). Furthermore, we have adopted the uniform expressions of the mass ratio Ag+/PVP in the revised manuscript.
- The fact that the addition of PVP allows the production of particles with smaller diameters and a more controlled particle size distribution was emphasized during the discussion. However, only one AgNPs sample was reported for the average size and relative particle size distribution histogram. Authors should provide this information for all samples prepared, at least in the Supporting Info section of the manuscript.
Response: Thanks for the referee's helpful suggestion. In the manuscript, the TEM images had reflected the synthesis results of AgNPs with different PVP levels, so we only supplied the particle size distribution histogram of AgNPs prepared in the optimal ratio of Ag+/PVP addition. Furthermore, the TEM images could better reflect the preparation effect of AgNPs.
- Pag 6 row 252. The comment refers to Figure 2a.
Response: Thanks for the referee's helpful suggestion. We have modified the comment to Figure 2a in Page 6 row 252.
- Pag 9 row 329. “Distilled water without any addition was taken as the blank” this sentence is unclear: I hope NaBH4 has been added to the blank. Please clarify.
Response: Thanks for the referee's helpful suggestion. We have modified the sentence as follows:
“Distilled water was taken as the blank group, 1,4-α-glucosidase was used for negative group and the reaction system in the blank and negative groups also included 4-NA and NaBH4.”
- In the first part of the manuscript, until page 9, particles obtained in the presence of PVP as stabiliser were called mAgNPs, whereas in the second part they were called AgNPsM. Authors should use consistent nomenclature.
Response: Thanks for the referee's helpful suggestion. We have modified and use consistent nomenclature “mAgNPs” in the revised manuscript.
- Pag 10 rows 384-388. This paragraph should be deleted.
Response: Thanks for the referee's helpful suggestion. We have deleted the paragraph in Page 10 rows 384-388.
- There are some typographical and grammatical errors in the text. I advise the authors to check the manuscript carefully.
Response: Thanks for the referee's helpful suggestion. Thanks for the referee's helpful suggestion. We are very grateful to the reviewer for pointing out the lack of the original manuscript. We have checked the manuscript carefully and then corrected the errors. Before we submit the revised manuscript, the English in this manuscript has been checked and modified by a native English speaker.
In addition, a number of typographical and language usage errors we noticed whilst revising the manuscript has been corrected. All changes are marked in red fonts throughout the revised manuscript.
We look forward to hearing from you again.
With kind regards,
Prof. Dr. Wen-Yong Lou
Lab of Applied Biocatalysis,
South China University of Technology,
Guangzhou 510640, China
Tel/Fax: +86-20-22236669
E-mail: wylou@scut.edu.cn (Prof. Lou)
Reviewer 3 Report
This manuscript deals with "1,4-α-glucosidase from Fusarium solani for controllable biosynthesis of silver nanoparticles and their multifunctional applications". This article claims that using of Fusarium solani- based silver nanoparticles could be a suitable for various applications. Therefore, I suggest a minor correction and require a detailed clarification. Correction to be addressed by the authors as follows: The abstract is not well organized, where the sentences are incomplete and no continuity is there. It would be feasible, if include the significance of the current study in the abstract. A brief description of how the authors selected information from the literature in the databases, as well as what time period they searched for, is missing.
Authors should justify and expand the information on the advantages of this method for biomedical applications.
Authors should specify the main experimental conditions used on the evidences from the literature. Where they briefly describe the most important data reported in the literature in a homogeneous manner and sequence reinforcing the relevance of this method as novel alternative.
Authors should discuss whether the use of this method represents a solid alternative to existing commercial agents .
Please add below studies to your manuscript in discussion section using below manuscripts:
DOI: 10.2217/nnm-2020-0441
DOI: 10.1016/j.jcis.2020.10.047
Conclusions should reaffirm the fundamental contribution of this paper.
Author Response
Dear Reviewer,
We are very grateful to you for your consideration of our manuscript (Manuscript ID: ijms-2215846) for publication in International Journal of Molecular Sciences and helpful suggestions for revision. We have revised the manuscript carefully according to the reviewers’ comments and the editor’s suggestions. The issues raised by the reviewers have been addressed as follows. Page, line and figure numbers refer to the revised manuscript.
Response to the third Reviewer’s comments
This manuscript deals with "1,4-α-glucosidase from Fusarium solani for controllable biosynthesis of silver nanoparticles and their multifunctional applications". This article claims that using of Fusarium solani-based silver nanoparticles could be a suitable for various applications. Therefore, I suggest a minor correction and require a detailed clarification. Correction to be addressed by the authors as follows:
- The abstract is not well organized, where the sentences are incomplete and no continuity is there. It would be feasible, if include the significance of the current study in the abstract. A brief description of how the authors selected information from the literature in the databases, as well as what time period they searched for, is missing.
Response: Thanks for the referee's helpful suggestion. We have re-organized the Abstract section.
- Authors should justify and expand the information on the advantages of this method for biomedical applications.
Response: Thanks for the referee's helpful suggestion. We have added the information on the advantages of this method for biomedical applications.
- Authors should specify the main experimental conditions used on the evidences from the literature. Where they briefly describe the most important data reported in the literature in a homogeneous manner and sequence reinforcing the relevance of this method as novel alternative.
Response: Thanks for the referee's helpful suggestion. In our manuscript, we supplied a green synthesis method to prepare AgNPs, and further excavated the 1,4-α-glucosidase from F. solani who was responsible for the controllable bio-synthesis of AgNPs with the help of PVP and β-NADPH. On the contrast, the specific preparation method of AgNPs appeared less important.
- Authors should discuss whether the use of this method represents a solid alternative to existing commercial agents.
Response: Thanks for the referee's helpful suggestion. Our method represents an alternative method with great potential for development, but now the main preparation technology of AgNPs was chemical synthesis.
The solid strategy to existing commercial agents
- Please add below studies to your manuscript in discussion section using below manuscripts:
DOI: 10.2217/nnm-2020-0441
DOI: 10.1016/j.jcis.2020.10.047
Response: Thanks for the referee's helpful suggestion. We have added the above two literatures in the results and discussion section of the revised manuscript.
- Conclusions should reaffirm the fundamental contribution of this paper.
Response: Thanks for the referee's helpful suggestion. We have re-written the Conclusions section.
In addition, a number of typographical and language usage errors we noticed whilst revising the manuscript has been corrected. All changes are marked in red fonts throughout the revised manuscript.
We look forward to hearing from you again.
With kind regards,
Prof. Dr. Wen-Yong Lou
Lab of Applied Biocatalysis,
South China University of Technology,
Guangzhou 510640, China
Tel/Fax: +86-20-22236669
E-mail: wylou@scut.edu.cn (Prof. Lou)
Round 2
Reviewer 1 Report
The answer to question 4 should be supplemented in the paper, not just in the response.
Author Response
Dear Reviewer,
We are very grateful to you for your consideration of our manuscript (Manuscript ID: ijms-2215846) for publication in International Journal of Molecular Sciences and helpful suggestions for revision. We have revised the manuscript carefully according to the reviewer’s comments. The issues raised by the reviewers have been addressed in the revised manuscript.
Response to the Reviewer’s comments
The answer to question 4 should be supplemented in the paper, not just in the response.
Response: Thanks for the referee's helpful suggestion. We have added the total number of particles particles in the captions of Figure 2c.
In addition, a number of typographical and language usage errors we noticed whilst revising the manuscript has been corrected. All changes are marked in red fonts throughout the revised manuscript.
We look forward to hearing from you again.
With kind regards,
Prof. Dr. Wen-Yong Lou
Lab of Applied Biocatalysis,
South China University of Technology,
Guangzhou 510640, China
Tel/Fax: +86-20-22236669
E-mail: wylou@scut.edu.cn (Prof. Lou)